# A Recipe for Success? A Nutrient Analysis of Recipes Promoted by Supermarkets

**DOI:** 10.3390/ijerph17114084

**Published:** 2020-06-08

**Authors:** Jasmin Wademan, Gael Myers, Anne Finch, Satvinder S. Dhaliwal, Jane Scott, Andrea Begley

**Affiliations:** 1School of Public Health, Curtin University, Perth, WA 6102, Australia; jasmin.wademan@postgrad.curtin.edu.au (J.W.); s.dhaliwal@curtin.edu.au (S.S.D.); jane.scott@curtin.edu.au (J.S.); 2Cancer Council Western Australia, Level 1, 420 Bagot Road, Subiaco, WA 6008, Australia; gmyers@cancerwa.asn.au (G.M.); afinch@cancerwa.asn.au (A.F.); 3Duke-NUS Medical School, National University of Singapore, 8 College Rd, Singapore 169857, Singapore

**Keywords:** cooking, content analysis, food magazine, nutrient analysis, public health, recipe magazine

## Abstract

Recipe use impacts eating habits, yet there is limited research investigating the nutritional quality of recipes. Supermarket recipe magazines command large readerships, with over 4 million readers for each of the two major Australian supermarket publications. Assessing the nutrient content of featured recipes is therefore of public health interest. The nutrient content of 312 main-meal recipes from *Coles*^®^
*Magazine* and *Woolworths Fresh*^®^ were analyzed and compared against a traffic-light system for classifying nutrients of concern in chronic disease. Nutrient content was compared across recipe type (standard, advertorial and celebrity) and between recipes with and without health or nutrient claims. Overall compliance with the traffic-light criteria was low, with less than half of recipes meeting the target. Advertorial recipes had a higher energy (*p* = 0.001), saturated fat (*p* = 0.045) and sodium (*p* ≤ 0.001) content per serve, and per 100 g for sodium (*p* ≤ 0.001) compared to standard and celebrity recipes. Recipes with claims had greater compliance to the nutrient criteria (*p* < 0.001) compared to those without. These findings support previous research highlighting the poor nutritional quality of published recipes from a variety of sources.

## 1. Introduction

A higher frequency of home cooking has been associated with improved diet quality [1,2]. Research in this area has primarily focused on cooking skills, with limited investigation of what and how people are cooking [2]. Evidence suggests that the use of food magazines and recipes impacts food purchasing and preparation habits, with consumers describing the use of recipe magazines as a way to “make over” their food habits [1]. However, available research investigating the nutritional quality of recipes in the public domain indicates that the nutrient composition of published recipes is in contradiction with public health recommendations [3,4,5,6,7,8,9].

An Australian study comparing the nutrient content of celebrity-chef cookbook recipes against the Victorian Healthy Choices Traffic-Light System [4] found that 68% of recipes were classified as “red”, indicating high energy, saturated fat, sugar and/or sodium content [3,4]. Likewise, investigation of the nutrient content of recipes promoted by English TV chefs found the majority to be high in fat, saturated fat, and sodium [5,6]. A study comparing the nutrient content of online recipes against the World Health Organization (WHO) nutrition guidelines found less than 1% of recipes to be fully compliant with the guidelines [7]. A comparison of recipes published on online food blogs found no difference in recipes promoted as healthy compared to standard recipes with respect to energy, carbohydrate, sugar and sodium content [8]. Recipes promoted as healthy were found to be higher in protein and fiber content; however, less than 10% of “healthy recipes” were compliant with WHO nutrition guidelines [9]. Only one study investigating the nutritional quality of recipe magazines was identified. This study, focusing on a popular Swedish publication, found that recipes contradicted national nutrition recommendations as they contained high levels of fat and sodium [8].

There is currently limited research in an Australian context, with only two studies identified. These have focused on celebrity-chef cookbook recipes [4] and food blogs [9], respectively. Food magazines are a popular source of cooking information and recipes in Australia [10]. The food magazine category is dominated by free recipe magazines produced and distributed by the two major Australian supermarket chains, Coles^®^ and Woolworths^®^. *Coles*^®^
*Magazine* reaches an audience of 4.8 million people (19.2% of the population) and Woolworths Fresh^®^ magazine reaches 4.2 million people (16.7% of the population) [11,12]. The production of supermarket magazines has been identified as a marketing strategy to create deeper connection to consumers and bring to life the brand image [13]. To our knowledge, no studies to date have investigated the nutritional quality of supermarket promoted recipes, which may differ in nutritional content to the recipes in previously researched publications as supermarket recipes are frequently promoted as quick, healthy and budget-friendly [14].

The huge reach and accessibility of supermarket recipe magazines, in addition to the documented influence of recipes on cooking and eating behaviors, indicates the potential for significant public health influence. This study aimed to assess the nutritional quality of recipes from supermarket magazines against a set of traffic-light criteria used and developed for a Government mass media healthy lifestyle campaign. A secondary objective was to compare the nutrient content of standard supermarket-produced recipes with advertorial and celebrity-chef recipes, as well as comparing the nutrient content of recipes with and without health or nutrient claims.

LiveLighter^®^ is a Western Australian social marketing campaign funded by the Western Australian Department of Health that encourages adults to eat well, be physically active, and maintain a healthy weight. The campaign commenced in June 2012 and is currently active as of May 2020. The campaign has run at various time points in other jurisdictions within Australia including Victoria, Northern Territory, South Australia, and Tasmania. In addition to elements such as mass media, research, and advocacy, evidence-based information and resources are developed and distributed to the population to facilitate behavior change. During the development of the campaign, a set of recipe guidelines, including nutrient criteria for nutrients of concern in chronic disease, were established [15]. These were developed based on a review of existing nutrient guidelines from previous public health campaigns [16,17], the Australian Dietary Guidelines [18] and Food Standards Australian and New Zealand recommendations [19]. Recipes included on the LiveLighter^®^ website must meet the nutrient criteria and recipe guidelines. In addition, the nutrient criteria are promoted to the public as a way to assess the nutritional quality of everyday foods consumed.

## 2. Materials and Methods

This project was a retrospective, cross-sectional analysis of the nutrient content and nutritional quality of recipes from Australian supermarket magazines.

### 2.1. Sampling

A complete list of all main-meal recipes was obtained from the online issues of *Coles*^®^
*Magazine* and *Woolworths Fresh*^®^ published between July 2018 and June 2019. Each magazine produces a monthly issue, except for a combined January/February issue for *Woolworths Fresh*^®^. The sampling frame was drawn from the index page and included all recipes defined as a main-meal by the magazine, which included recipes by celebrity-chefs. The two magazines differed in the inclusion of advertorial recipes in the index list; *Coles*^®^
*Magazine* included these in the index list while *Woolworths Fresh*^®^ did not. Hence, *Woolworths Fresh*^®^ was manually searched for additional advertorial main-meal recipes. The total number of main-meal recipes available across the time period for the two magazine types was 641, with the number of recipes in any one issue varying from 13 to 41. A sample of 13 recipes per issue was selected as this was the minimum number of recipes within a single issue. A simple random sampling technique was used, using an online random number generator (researchrandomizer.org), to select 13 main-meal recipes from each magazine. A double sample of 26 recipes was taken from the January/February issue of *Woolworths Fresh*^®^, giving a total sample size of 312 recipes to be included in the analysis.

### 2.2. Nutrient Analysis

The nutrient content of 312 main-meal recipes was analyzed using Foodworks 9 (Xyris Software Pty Ltd., Spring Hill, Queensland, Australia), using the Australian food composition databases AUSFOODS2017, AUSBRANDS2017 and NUTTAB2010 [20]. Foodworks was used to calculate energy, protein, total fat, saturated fat, sugar, sodium, and dietary fiber content per serving and per 100 g. Additional information was collected on recipe type (standard, advertorial or celebrity) and the presence of “claims”. “Claims” were defined as any health, nutrient, food group or dietary requirement (e.g., gluten-free) claim made by the magazine. Recipe type was defined as the source of the recipe, being either magazine-produced (standard), paid advertising (advertorial) or developed by a celebrity-chef (celebrity).

When entering recipes into Foodworks, ingredients were entered in their cooked state, where appropriate, with conversion factors applied to adjust for differences in cooked and raw weight [21]. The cooked state was used, rather than the raw ingredient as listed in the recipe, to ensure the analyzed nutrient content best matched the meal as consumed. Only food items listed in the ingredients section were included; any additional items (such as salt or oil) mentioned in the method but not included in the ingredients list were not included in the analysis as insufficient detail was provided in the methods section on the quantity of the ingredient to use. Soup recipes and recipes with greater than 1 liter of added liquid were adjusted for water loss [22]. Where an exact ingredient was not available, a suitable alternative was chosen from the Foodworks list. Additional ingredient information such as the drained weight of jarred foods was obtained from the supermarket websites as required (shop.coles.com.au; woolworths.com.au).

Recipes were entered into Foodworks by one author (JW), a student dietitian. A second author (AF), a dietitian experienced in the use of the Foodworks software, analyzed a random sub-sample of 20 recipes to determine inter-rater reliability. An acceptable percentage similarity was set at 85% [23]. Comparison of the data for all nutrients per serve and per 100 g found an overall average percentage similarity of 89.8%, ranging from 85.0–93.5% similarity for individual nutrients, this degree of similarity was therefore deemed acceptable.

### 2.3. Promotional Foods Content Analysis

Additional information was collected on the types of foods being promoted in sampled advertorial recipes. Promoted foods were classified into groups as outlined in Charlton et al.’s method; core foods (fruit, vegetables, breads and cereals, dairy products and alternatives, meats and alternatives), unsaturated fats and oils, discretionary foods (energy-dense and nutrient-poor) and non-food items [24]. The definitions of core foods and discretionary foods were taken from the Australian Dietary Guidelines [18].

### 2.4. LiveLighter^®^ Traffic-Light Criteria

To assess nutritional quality, recipes were assigned a traffic-light color for total fat, saturated fat, sodium and dietary fiber based on the LiveLighter^®^ traffic-light criteria [15]. Table 1 outlines the criteria for classification of each nutrient. The sugar criterion was excluded from this analysis as all recipes were main meals and therefore did not contain substantial amounts of sugar. Overall compliance with the LiveLighter^®^ criteria was assessed, with recipes being compliant if they had a minimum of two nutrients classified as green and had no nutrients classified as red, as per the LiveLighter^®^ recipe guidelines [15]. The overall assessment of compliance excludes dietary fiber due to the differences in the way this nutrient is categorized in the LiveLighter^®^ traffic-light criteria (i.e., per serve and no amber rating) [15].

### 2.5. Statistical Analysis

Statistical analysis was conducted usingStatistical Package for the Social Sciences SPSS, Version 25 (IBM Corporation, Armonk, NY, USA). Descriptive analysis was conducted for all data. All continuous data were assessed for normality using skewness, kurtosis, and normality plots. Independent t-tests and Mann-Whitney U tests were used to compare the nutrient content of recipes with and without health or nutrition claims. One-way analysis of variance ANOVA and Kruskal-Wallis tests were used to compare the nutrient content between recipe types, with least significant difference LSD used for post-hoc analysis. Chi-square and Fisher’s exact tests were used to compare differences in the traffic-light distribution of nutrients as per the LiveLighter^®^ criteria. The level of statistical significance was set at *p* < 0.05 for all tests.

## 3. Results

A total of 312 main-meal recipes were included in this study (156 from each of the included supermarket publications). With respect to recipe type, there were a total of 234 standard recipes (75% of the sample), 42 advertorial recipes (13% of the sample), and 36 celebrity-chef recipes (12% of the sample). Health or nutrition claims were present on 75 of the sampled recipes (24% of the sample), while a further 237 recipes did not have claims (76% of the sample). Claims included health and nutrient claims such as “low fat” and food preference or dietary requirement claims such as “gluten-free”.

### 3.1. Nutrient Content

The mean nutrient content and serve size of recipes is presented in Table 2 with comparisons across recipe type and between recipes with and without claims. Advertorial recipes had a significantly higher serve size (*p* = 0.002) and energy (*p* = 0.001), saturated fat (*p* = 0.045) and sodium (*p* < 0.001) content per serve when compared to standard and celebrity recipes (Table 2). A significant difference in sodium content per 100 g was observed across all three recipe type categories (*p* < 0.001), with advertorial recipes having the highest mean content (272 mg/100 g) (Table 2). Recipes with a nutrient or health claim had a significantly smaller serve size (*p* = 0.001) and were significantly lower in energy (*p* < 0.001), fat (*p* = 0.001), saturated fat (*p* < 0.001) and sodium (*p* < 0.001) content per serve, compared to recipes without claims (Table 2). Similarly, recipes with claims were significantly lower in energy (*p* = 0.002), saturated fat (*p* < 0.001), and sodium (*p* < 0.001) content per 100 g compared to those without claims (Table 2).

### 3.2. Compliance with LiveLighter^®^ Traffic-Light Criteria

Figure 1 presents the proportion of recipes classified as red, amber or green for each nutrient of interest (using the LiveLighter^®^ criteria (see Table 1)). There were significant differences in the overall distribution of traffic-light colors for celebrity recipes compared to promotional or standard recipes for total fat (*p* = 0.047) and across all recipe types for sodium (*p* = 0.001). Celebrity recipes were more likely to be classified as red for total fat (36.1%), compared to standard recipes (18.8%; *p* = 0.033) and advertorial recipes (11.9%; *p* = 0.015). Advertorial recipes were more likely to be in the red category and less likely to be in the green category for sodium when compared to other recipe types (Figure 1). Comparing recipes with and without claims, a significant difference was observed in the traffic-light distribution for saturated fat (*p* = 0.003) and sodium (*p* < 0.001) (Figure 1), with recipes making health or nutrition claims more likely to be in the green category and less likely to be in the red category for both nutrients. Most recipes across all category types were classified as green for fiber content per serve (Figure 1). An alternative fiber category cut-off of 5 g has been proposed [25]. If this stricter criterion is applied the proportion of total recipes classified as green for fiber content would reduce from 92.3% to 79.5% (Figure 1, Table A1).

Across the total sample, 43.3% of recipes were compliant with the overall requirement to have a minimum of two green nutrients and no red nutrients (Table 3). No significant differences were observed when comparing overall compliance across recipe types; however, recipes with claims were significantly more compliant than those without (61.3% vs. 38.7%, *p* < 0.001) (Table 3).

### 3.3. Promoted Foods Content Analysis

Sixty-four food products and two non-food products were promoted in the 42 advertorial recipes. Of the promoted food items, 33.3% were classified as core foods, 12.1% as unsaturated spreads and oils and 51.5% as discretionary items (Table 4).

## 4. Discussion

The purpose of this study was to assess the nutrient content of main-meal recipes included in Australian supermarket recipe magazines and compare this against a measure of nutritional quality, the LiveLighter^®^ traffic-light criteria. To our knowledge, this is the first study investigating supermarket recipe magazines and the first Australian study to analyze a range of nutrients in magazine recipes.

This study found that advertorial recipes contained more energy, saturated fat, and sodium per serve when compared to standard and celebrity recipes. This result is likely primarily due to the significantly greater serving size of advertorial recipes, as only sodium differed significantly when comparing per 100 g. The larger serving size may be explained by the financial motive of such recipes. Advertorial recipes are designed to sell a food product and advertising larger portions may be a strategy for increased sales, as is seen in fast food marketing strategies [26]. However, the significant findings from the per serve analysis should not be disregarded as this is likely the portion size that individuals consume. Consumption of larger meal sizes has been associated with excess energy intake and weight gain and hence may be a contributing factor to obesity and other chronic illness [27,28,29].

The analysis of products advertised within advertorial recipes revealed that the majority were unhealthy “discretionary” food items, while a smaller proportion were healthy “core” foods. These findings are in line with content analysis studies of supermarket catalogues, identifying that the foods advertised by supermarkets are predominantly ultra-processed, discretionary items [30]. A previous analysis of Coles^®^ and Woolworths^®^ catalogues revealed a ratio of core to discretionary foods of 0.8:7 [24]. The findings from the present study suggest that the tendency for supermarkets to promote discretionary foods follows through into supermarket recipe magazines.

Celebrity recipes were found to be less compliant with the LiveLighter^®^ nutrient criteria for total fat, while advertorial recipes were less compliant with sodium guidelines. Furthermore, advertorial recipes were less compliant with the overall LiveLighter^®^ recommendation to have a minimum of two nutrients in the “green” range and no nutrients in the “red” range. The increased portion size, significantly greater sodium content, high proportion of discretionary foods advertised and lower compliance with LiveLighter^®^ nutrient criteria highlights the need to clearly mark advertorial recipes to provide consumers with adequate information to make informed dietary choices.

The present study found that recipes with a health or nutrient claim were significantly lower in energy, saturated fat, and sodium content per serving and per 100 g. Recipes making a claim were also found to be significantly more compliant with the LiveLighter^®^ nutrient criteria than those without. These findings suggest that recipes making a claim are a better choice than those without a claim. It should be noted, however, that over one third of recipes with claims were still found to be non-compliant with LiveLighter^®^ nutrient criteria. Interestingly, these results are in contradiction with other research and may not be transferable to other recipe sources [31,32]. Previous research comparing recipes promoted as healthy against control recipes found that there were limited differences between the two recipe types in their nutritional quality [4,9,31,32]. Further analysis including investigation of individual claims, for example investigation of whether claims are supported by an analysis of actual nutrient content, should therefore be conducted.

Overall a small proportion of the recipes included in this study met LiveLighter^®^ nutrient criteria for nutrients of interest in chronic disease development (total fat, saturated fat, and sodium). Similarly, most recipes did not comply with the overall LiveLighter^®^ nutrient guidelines. This is consistent with other research on publicly available recipes that has found the majority to be of poor nutritional quality [4,5,6,7,8,9]. While this is the only study to use the LiveLighter^®^ criteria for analysis, other studies have found similarly low compliance with other nutrient criteria. Trattner et al. found that only 0.14% of online recipes were fully compliant with the WHO criteria [7]. Similarly, other research has found published recipes to be high in fat, saturated fat, and sodium [5,6,7,8]. Other studies assessing the nutrient content of recipes published online against traffic-light systems have also found a low proportion of recipes to be in the “green” range for fat, saturated fat, and sodium [4,7].

Dietary fiber was the only nutrient to obtain a favorable rating with the LiveLighter^®^ criteria, with most recipes classified as green. However, the LiveLighter^®^ criteria for fiber is relatively low with less than 3 g per serve considered low fiber and more than 3 g per serve considered acceptable. Considering that the recipes included in this study were all main meals, a contribution of greater than 3 g of dietary fiber would likely be required to meet the daily adequate fiber intake of 25 g for women and 30 g for men [32]. Similar studies assessing the nutrient content of recipes against alternative nutrient criteria have used higher cut-offs, for example 5 g in the study by Silva et al. [25]. If this 5 g cut-off were applied, only 79.5% of recipes would have been classed in the green category. This indicates that the LiveLighter^®^ criterion for dietary fiber may be too low for the purpose of assessing acceptable levels in main meals.

The present study has several limitations. First, data collection did not include ingredients such as seasoning and oil that were mentioned in the method section but not the ingredient list of the recipe. This means that the results may have underestimated the fat and sodium content of the recipes and overestimated the number of recipes meeting the LiveLighter^®^ criteria for these nutrients. Secondly, only the main-meal recipes were included in this analysis and so it is unclear the extent to which other recipes included in supermarket recipe magazines, such as desserts and light meals, adhere to the LiveLighter^®^ nutrient criteria. Lastly, a decision was made not to include sugar in the assessment of overall compliance with the LiveLighter^®^ nutrient criteria as all recipes analyzed in this study were main-meal recipes and therefore were not high in sugar. As the recipes were still required to have a minimum of two nutrients in the green category the number of recipes conforming to the overall traffic-light criteria may have been lower than if sugar was included in the assessment, as the majority of recipes would then have had one nutrient classified in the green category automatically. Finally, although the proportion of the Australian population accessing supermarket magazines is high, there is no known data on how people are using these magazines i.e., how many readers are actually cooking recipes from these magazines. Hence, further research is required to properly assess the public health influence of these magazines.

This study focused on the key nutrients of interest in chronic disease risk, in line with the LiveLighter^®^ nutrient criteria. Future research could build on this by including additional nutrients or using broader nutrient criteria such as the WHO criteria used in other studies, making results more comparable to the body of research. Assessment of a wider range of recipe types, such as appetizer and dessert recipes, would also be beneficial, as well as a content analysis of supermarket magazines, e.g., the ratio of advertorial to non-advertorial recipes. Future research could also compare supermarket recipe magazines with other recipe sources such as subscription magazines and online recipe websites to identify the best choice when recommending recipes to the public.

## 5. Conclusions

This research indicates that recipes promoted by Australian supermarkets are of poor nutritional quality when assessed against a set of traffic-light nutrient criteria for total fat, saturated fat, and sodium. The serving size of advertorial recipes was significantly larger and the sodium, fat, and energy content per serve was significantly higher when compared to standard and celebrity recipes. Advertorial recipes also promoted the use of unhealthy discretionary foods and were less likely to be compliant with the traffic-light criteria compared to other recipe types. Recipe magazines should therefore be used with caution and advertorial recipes should be clearly marked to allow the public to make informed choices about the meals they prepare and consume.

## Figures and Tables

**Figure 1 ijerph-17-04084-f001:**
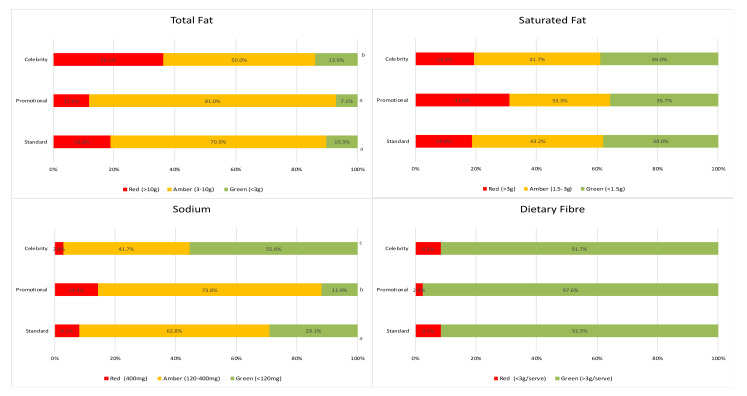
Proportion of recipes in each LiveLighter^®^ traffic-light category for individual nutrients and comparison (i) across recipe type and (ii) between recipes with and without claims. ^a,b,c^ Values within each analysis category with different superscript letters were significantly different from each other (*p* < 0.05).

**Table 1 ijerph-17-04084-t001:** LiveLighter^®^ traffic-light criteria.

Nutrient	Green	Amber	Red
	“Best Choice”	“Okay Sometimes”	“Poor Choice”
Total Fat (per 100 g)	<3 g	3–10 g	>10 g
Saturated Fat (per 100 g)	<1.5 g	1.5–3 g	>3 g
Sugar (per 100 g)	<5 g	5–15 g	>15 g
Sodium (per 100 g)	<120 mg	120–400 mg	>400 mg
Fiber per serve	>3 g	-	<3 g

Adapted from LiveLighter^®^ recipe guidelines [15].

**Table 2 ijerph-17-04084-t002:** Mean (± SD) serve size, energy, macronutrient and sodium content per serve and per 100 g and comparison (i) across recipe type and (ii) between recipes with and without claims.

Per Serving										Per 100 g						
		Recipe Type			*p*	Claims		*p*		Recipe Type			*p*	Claims		*p*
	Total (*n* = 312)	Standard (*n* = 234)	Advertorial (*n* = 42)	Celebrity (*n* = 36)		Yes (*n* = 75)	No (*n* = 237)		Total (*n* = 312)	Standard (*n* = 234)	Advertorial (*n* = 42)	Celebrity (*n* = 36)		Yes (*n* = 75)	No (*n* = 237)	
Serve size (g)	415 ± 138	403 ± 126 ^a^	485 ± 154 ^b^	414 ± 168 ^a^	0.002	374 ± 107	428 ± 144	0.001	-	-	-	-	-	-	-	-
Energy (kJ)	2439 ± 908	2352 ± 907 ^a^	2896 ± 835 ^b^	2472 ± 852 ^a^	0.001	1997 ± 630	2579 ± 938	<0.001	609 ± 185	600 ± 184	621 ± 150	654 ± 222	0.241	552 ± 168	627 ± 187	0.002
Energy (kCal)	583 ± 217	562 ± 217 ^a^	692 ± 200 ^b^	591 ± 204 ^a^	0.001	477 ± 151	616 ± 224	<0.001	146 ± 44	143 ± 44	148 ± 36	156 ± 53	0.241	132 ± 40	150 ± 45	0.002
Protein (g)	35.7 ± 17.8	35.1 ± 17.8	38.1 ± 14.6	37.0 ± 21.2	0.228	30.8 ± 15.3	37.3 ± 18.3	0.004	9.1 ± 4.2	9.1 ± 4.3	8.1 ± 2.7	9.9 ± 4.8	0.338	8.8 ± 4.6	9.2 ± 4.1	0.339
Fat (g)	28.2 ± 13.7	27.4 ± 13.6	31.7 ± 11.7	29.7 ± 16.0	0.144	23.6 ± 11.4	29.7 ± 14.0	0.001	7.3 ± 3.7	7.2 ± 3.6	7.0 ± 2.8	8.4 ± 5.1	0.730	6.7 ± 3.7	7.5 ± 3.7	0.049
Saturated Fat (g)	8.6 ± 5.9	8.4 ± 5.8 ^a^	10.4 ±5.9 ^b^	8.2 ± 6.5 ^a^	0.045	5.8 ±3.7	9.5 ± 6.2	<0.001	2.2 ± 1.4	2.1 ± 1.4	2.3 ± 1.3	2.3 ± 2.0	0.582	1.6 ± 1.0	2.3 ± 1.5	<0.001
Sugar (g)	10.9 ± 6.8	10.8 ± 6.89	12.5 ± 6.89	9.5 ± 6.0	0.115	10.4 ± 6.8	11.0 ± 6.8	0.392	2.7 ± 1.6	2.7 ± 1.6	2.6 ± 1.3	2.5 ± 1.7	0.521	2.8 ± 1.8	2.6 ± 1.5	0.663
Sodium (mg)	838 ± 539	807 ± 509 ^a^	1241 ± 545 ^b^	569 ± 479 ^c^	0.000	577 ± 429	920 ± 544	<0.001	209 ± 135	208 ± 135 ^a^	272 ± 129 ^b^	143 ± 113 ^c^	<0.001	159 ± 119	225 ± 137	<0.001
Fiber (g)	7.9 ± 3.8	7.8 ± 3.7	8.2 ± 3.0	7.9 ± 4.9	0.367	7.5 ± 3.8	8.0 ± 3.8	0.340	1.9 ± 0.7	1.9 ± 0.7	1.8 ± 0.5	1.9 ± 1.0	0.367	2.0 ± 0.8	1.9 ± 0.7	0.525

^a,b,c^ Values within each recipe category with different superscript letters were significantly different (*p* < 0.05).

**Table 3 ijerph-17-04084-t003:** Overall compliance of recipes with LiveLighter^®^ nutrient criteria and comparison (i) across recipe type and (ii) between recipes with and without claims.

	Classification							
		Recipe Type			*p*	Claims		*p*
	Total (*n* = 312)	Standard (*n* = 234)	Advertorial (*n* = 42)	Celebrity (*n* = 36)		Yes (*n* = 75)	No (*n* = 237)	
Compliant *	43.3%	43.6%	33.3%	52.8%	0.222	61.3% ^a^	37.6% ^b^	<0.001
Non-compliant	56.7%	56.4%	66.7%	47.2%	0.222	38.7% ^a^	62.4% ^b^	<0.001

^a,b^ Values within each analysis category with unlike superscript letters were significantly different from each other (*p* < 0.05). * Compliant recipes must contain a minimum of two nutrient classified as green and no nutrients classified as red as per the LiveLighter^®^ criteria for fat, saturated fat, and sodium.

**Table 4 ijerph-17-04084-t004:** Promoted food items in sampled advertorial recipes.

	Classification				
	Core Foods ^a^	Unsaturated Spreads and Oils	Discretionary ^b^	Non-Food Items	Total
Frequency	22	8	34	2	66
Percentage	33.3%	12.1%	51.5%	3.0%	100%

^a^ Foods in one of the five core food groups as defined by the Australian Dietary Guidelines (fruit, vegetables, grains and cereals, meat and alternatives, or dairy and alternatives) [18]. ^b^ Foods not necessary to provide nutrients to the body and/or high in, saturated fat, added sugar or sodium as defined by the Australian Dietary Guidelines [18].

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
