# Peer review of "A Recipe for Success? A Nutrient Analysis of Recipes Promoted by Supermarkets"

_ijerph, 2020, doi:10.3390/ijerph17114084_

Round 1

Reviewer 1 Report

This original paper deals with an important issue which is on the WHO agenda of health topics deserving further scientific attention. The study comprises of an interesting and welcomed analysis of how/to what degree recipes circulated in popular magazines do not adhere to guidelines recommendations regarding healthy eating behaviours.

This research is well done and the results are well structured and presented.

I have only one comment regarding the Results section (table 2): I believe that if the authors had to present the results by means (probably the data distribution was normal), they also have to mention what the values after +/- represent (probably they are SD)? If the data distribution was not normal, the authors should have presented the results by medians and IQR.

Author Response

Thank you for your comment regarding the Results section (table 2). The analysis presented in Table 2 is correct. The distribution of the data was checked for normality, and hence parametric tests were used.  In cases when there was a slight doubt, an equivalent non-parametric test was conducted in addition, to confirm that the results of non-parametric hypothesis tests were similar to that of the parametric tests. Presenting results as mean ± standard deviation in Table 2, makes it easier to interpret of the outcome of the analyses.

Reviewer 2 Report

Although an important topic is addressed by the authors I feel that the sampling frame was limited as only supermarket magazines were used. The authors don’t explain the use of only this type of magazine except that people are likely more exposed to it. But exposure does not determine recipe use. Possibly providing amount of readers of these magazines that use recipes , of this data is known. I don’t think I missed this in the manuscript. 
The criteria used to determine quality of the recipes is one the authors appear to be familiar with but they do not provide adequate information about how this rating equates to chronic disease. The recipe types are provided but not explained. For example, what does the standard recipe type consist of, and what were the criteria?

Author Response

Thank you for the comment about the use of supermarket recipes, we have added into the limitation that there is no data on how consumers use these magazines such as using specific recipes. 

With regard to the comment about quality, in the Introduction line 79 to 85 the background to the quality criteria is provided in the introduction indicates that the nutrient criteria for nutrients of concern in chronic disease.

The recipe types are explained in Section 2.2 Nutrient analysis

Reviewer 3 Report

This is a very interesting paper, addressing a real-life influencer on people's diets.

The figures are not the most elegant ones I have ever seen. However, Fig 1 is okay, but the data presented in figure 2 could easily be presented in a table.

I struggle a bit with table 2. If I go on the assumption that the 'per serve' information and the 'per 100 g' information all came from the same saw data, then the 'per 100 g' information is directly proportional to the 'per serve' information. This in turn would mean that the percentage of the standard deviations (%SD) should be the same. However, the %SD are different. For instance the energy info (in kJ) for a standard recipe is reported as 2439 +/- 908 (per serve) and 609 +/- 185 per 100 g. The %SD are 37.2 and 30.4 for the 'per serve' and 'per 100 g' respectively. This strikes me as odd, since it all comes from the same raw data, and regardless of the ratios - the % should not change.

I randomly checked  a couple of other 'similar' data and similar discrepancies occur.

Author Response

Thank you for your comments.  We have changed Figure 2 to a table. 

With regard to Table 2, the reason SD values are not proportional is that there is no standard size for serving size.  The serving size of each recipe was calculated based on overall recipe weight divided by how many people the recipe was designed to serve. Therefore the proportion of per serve in relation to the 100 g information will be different.

Round 2

Reviewer 2 Report

Thank you to the authors for considering the recommendations on their manuscript. I thank them for updating the limitations of the study. And appreciate the authors pointing out areas in the manuscript that were overlooked in my comments. The manuscript contributes to the literature and provides an important information on the intersection between nutrition education and consumer exposure to nutrition.